# Prediction of Buckwheat Maturity in UAV-RGB Images Based on Recursive Feature Elimination Cross-Validation: A Case Study in Jinzhong, Northern China

**DOI:** 10.3390/plants11233257

**Published:** 2022-11-27

**Authors:** Jinlong Wu, Decong Zheng, Zhiming Wu, Haiyan Song, Xiaoxiang Zhang

**Affiliations:** 1College of Agricultural Engineering, Shanxi Agricultural University, Jinzhong 030801, China; 2College of Information Science and Engineering, Shanxi Agricultural University, Jinzhong 030801, China

**Keywords:** recursive feature elimination cross-validation, UAV-RGB images, prediction of buckwheat maturity, feature vectors, regression models

## Abstract

Buckwheat is an important minor grain crop with medicinal and edible functions. The accurate judgment of buckwheat maturity is beneficial to reduce harvest losses and improve yield. With the rapid development of unmanned aerial vehicle (UAV) technology, it has been widely used to predict the maturity of agricultural products. This paper proposed a method using recursive feature elimination cross-validation (RFECV) combined with multiple regression models to predict the maturity of buckwheat in UAV-RGB images. The images were captured in the buckwheat experimental field of Shanxi Agricultural University in Jinzhong, Northern China, from September to October in 2021. The variety was sweet buckwheat of “Jinqiao No. 1”. In order to deeply mine the feature vectors that highly correlated with the prediction of buckwheat maturity, 22 dimensional features with 5 vegetation indexes, 9 color features, and 8 texture features of buckwheat were selected initially. The RFECV method was adopted to obtain the optimal feature vector dimensions and combinations with six regression models of decision tree regression, linear regression, random forest regression, AdaBoost regression, gradient lifting regression, and extreme random tree regression. The coefficient of determination ($R^{2}$) and root mean square error (RMSE) were used to analyze the different combinations of the six regression models with different feature spaces. The experimental results show that the single vegetation index performed poorly in the prediction of buckwheat maturity; the prediction result of feature space “5” combined with the gradient lifting regression model performed the best; and the $R^{2}$ and RMSE were 0.981 and 1.70 respectively. The research results can provide an important theoretical basis for the prediction of the regional maturity of crops.

## 1. Introduction

Buckwheat is an important minor grain crop in China. It is helpful to lower blood pressure, control diabetes, and improve digestion and cholesterol level [1]. The harvest time of the crop has a significant impact on its yield and quality; early or late harvest is not conducive to the high yield and income of the crop [2]. The biggest characteristic of buckwheat during its growth is the long overlapping period, in which the phenotypic characteristics of buckwheat change greatly. The overall growth period can be divided into four shorter sub-periods: budding, flowering, growth, and maturity. Therefore, the prediction of buckwheat maturity can be realized according to the phenotypic characteristics in these different periods. The accurate judgment of buckwheat maturity is beneficial to the advance scheduling of harvesters, and can effectively reduce harvest losses and improve yield. The traditional methods for evaluating buckwheat maturity are mainly based on field measurement and the experience of farmers. In large-scale planting, subjective misjudgment, manpower, and material resources waste are all factors. In recent years, the development of UAV has provided a new idea for the prediction of crop maturity, and the UAV technology can be used to further explore the features highly related to buckwheat planting information. In the flowering period, grains with different maturity levels will appear on the same buckwheat plant, and plants with different maturity levels will also appear on the same plot. If the harvest time is too early, most of the grains will be immature, and if the harvest time is too late, the grains will fall off, which will cause great economic losses. Therefore, accurate calibration of buckwheat maturity is of great significance to ensure buckwheat yield. Generally, when the grain maturity of a single buckwheat plant reaches 75–80%, the color of the grain turns brown or gray [3], which means that the single buckwheat plant is mature. Since UAV photography can only obtain the canopy image of buckwheat, it cannot accurately describe the mature state of each buckwheat plant, and then the overall maturity of each buckwheat plot can be calibrated. The empirically based maturation calibration process allows an error of one to two days.

UAV technology has been widely used in crop growth monitoring [4,5], pest and disease control [6], soil analysis and planning [7,8], precision fertilization [9], and other aspects, and the prediction of crop maturity is a major application of crop growth monitoring. At present, there have been studies on UAV remote sensing platforms to predict the maturity of agricultural products. Rodrigo Trevisan et al. [10] used convolutional neural networks (CNN) to predict the maturity of soybean in airborne RGB images, and the prediction results of RMSE can reach 2.0 days. Jing Zhou et al. [11] predicted the maturity of soybean in airborne multispectral images by partial least square regression (PLSR), and the prediction results of $R^{2}$ and RMSE were 0.81 and 1.4 days, respectively. Neil Yu et al. [12] developed a dual-camera, high-throughput phenotype platform mounted on a UAV and used the random forest method to measure soybean maturity; the prediction accuracy was 93%. However, most of the above research is aimed at the prediction of maturity in UAV images such as soybeans and wheat, and research on maturity prediction of minor grain crops such as buckwheat are fewer. Generally, methods of judging buckwheat maturity are mainly based on its phenotypic characteristics. At present, research on crop phenotype in UAV images primarily focus on indexes of leaf area index (LAI), leaf dry matter (LDM) [13], plant density (PD) [14], yield prediction [15], and above ground biomass (AGB) [16]. At the same time, most of the crop phenotype research based on UAV consists of multispectral and hyperspectral images. Experimental results show that the inversion of physical and chemical parameters can be realized by using the strong correlation between the fixed bands of multispectral and hyperspectral spectra and the biochemical moisture and pigment of plants. Since buckwheat is widely planted in China, and the UAVs equipped with multispectral and hyperspectral cameras are expensive, the data processing is more complicated, making it difficult to apply them to practical production. UAVs equipped with an RGB camera have been widely used in the field of crop classification and recognition due to the camera’s low cost and easy access. Hence, this study deeply explores the potential value of UAV-RGB images of buckwheat for maturity prediction by extracting multiple feature vectors from the images, obtaining the optimal combination of feature vectors and the optimal regression model, and demonstrating the prediction of buckwheat maturity in UAV-RGB images.

In this study, a method for buckwheat maturity prediction was proposed based on easily accessible and low-cost high-resolution UAV-RGB images. The main contributions are as follows:(1)The UAV-RGB images of buckwheat are collected periodically in the overlapping period, and then the vegetation indexes, color and texture features are extracted.(2)In view of interference information, such as bare ground in the images, and the subjectivity of feature selection in current research, correlation analysis and recursive feature elimination cross-validation methods are adopted for feature selection. Combining the selected features with multiple regression models allows the optimal combination of feature vectors and the optimal regression prediction model to be determined.(3)We evaluate the accuracy of the prediction model proposed in this paper in order to provide reference for UAV remote sensing detection for crops.

## 2. Materials and Methods

### 2.1. Research Area

The experiment was carried out in 2021 in the buckwheat research experimental field (37°26′2.4″ N−37°26′6″ N, 112°35′34.8″−112°35′42.0″ E) of Shanxi Agricultural University in Shenfeng village, Taigu District, Jinzhong City, Shanxi Province. The sowing time of buckwheat was July 2, and the variety was sweet buckwheat “Jinqiao No. 1”. During the experiment, the highest temperature was 31 degrees, the lowest temperature was 6 degrees, and the average rainfall was 119.25 mm. The total area of the experimental field was 13,021 m^2^, which was divided into 35 experimental plots with an average area of 3 × 3 m for each plot (Figure 1).

### 2.2. UAV Images Acquisition

The aerial image acquisition equipment is Dajiang PHANTOM 4RTK UAV (equipped with 20-megapixel 1-inch CMOS sensor), which is carried with the RTK module, compatible with high-precision GNSS mobile stations, and has centimeter level positioning capability. Setting of aerial photography parameters: the flight height is 30 m, the flight speed is 2.6 m/s, the GSD is 0.82 cm/pixel, the heading overlap rate is 80%, and the side overlap rate is 70%. The collection time is mainly after the flowering period of buckwheat, and the specific collection dates are September 9, 15, 21, 28 and October 8, 12 in 2021, respectively. Figure 2 shows the images collected at the same position of the buckwheat experimental field in different periods. Figure 2a,b show the squaring period of buckwheat, during which the crop sizes are small and the crop rows are clearly visible. Figure 2c,d show the flowering period of buckwheat, during which buckwheat grows rapidly, the canopy is mainly the color of flowers, and the crop rows are gradually less obvious. Figure 2e,f show the growth period and maturity period of buckwheat, respectively, and it is clearly evident that there are mature grains on the crops, and the overall color of the canopy has started to darken and turn brown. The results show that buckwheat plots at the same location in different periods have obvious differences in canopy color and texture. Therefore, the initial feature vector can be constructed from the color and texture features of buckwheat plots to predict the maturity of buckwheat.

### 2.3. Image Segmentation

In the growth of buckwheat, it is difficult to achieve the segmentation of single buckwheat because of the high planting density and the serious overlapping of leaves. Therefore, image segmentation can be used to separate the canopy area from the background area. Since there are great color differences between the crop area and the background, the canopy area can be segmented by color features. After analysis, the image segmentation results of the original R, G, and B channels are not ideal, so the commonly used vegetation indexes [17] (VI) in UAV remote sensing are introduced for image segmentation. Through experimental comparison, the excess green index (ExG) has better segmentation performance; its formula is shown in (1):(1)ExG= 2G−R−B

In order to realize the segmentation of the buckwheat canopy, the original RGB image was converted into an ExG gray-scale image, and then the K-means algorithm was used to cluster the image pixels into buckwheat canopy and background (“0” represents the background, “1” represents the canopy). Then, morphological open and close operations [18] were applied for noise removal and holes filling, and the final binary image was the segmentation result of buckwheat canopy.

### 2.4. Feature Extraction

This study constructed prediction features from three aspects: vegetation indexes, color features, and texture features. As the research object in this paper is the UAV-RGB image, five vegetation indexes based on the R, G, and B bands were selected, including the normalized green red difference index (NGRDI) [19], Green Leaf Algorithm (GLA), Visible Atmospherically Resistant Index (VARI), ExG, and Normalized Difference Yellow Index (NDYI), and the corresponding formulas are shown in Table 1.

The UAV-RGB images acquired at the same time and location have great differences due to the influence of illumination. Therefore, the RGB images can be converted into multiple color spaces to reduce the influence of illumination on the prediction results. By analyzing the common color space HSV [20], HLS [21], and Lab [22], color feature vectors that have strong correlation with the prediction results of buckwheat maturity were explored. In the test, HSV_H, HSV_S, and HSV_V represent the H, S, and V components of the HSV color space, respectively; Lab_L, Lab_a, and Lab_b represent the L, a, and b components of the Lab color space, respectively; and HLS_H, HLS_L, and HLS_S represent the H, L, and S components of the HLS color space, respectively. Thus, nine color features of HSV, HLS, and Lab color spaces were selected for the alternative feature space used for the prediction of buckwheat maturity.

As the buckwheat rows in the squaring period, the shape of the flowers in the flowering period, the number of grains in the growth period, and the maturity period of the UAV-RGB image are quite different in texture, texture features can effectively predict the maturity of buckwheat. In this paper, eight common texture features of each buckwheat plot were selected for feature space construction. Including six gray-level co-occurrence matrix values [23], with local binary pattern (LBP) [24] and Gabor [25] texture feature mean among them, the gray-level co-occurrence matrix values are Homogeneity (HOM), Contrast (CON), Dissimilarity (DIS), Entropy (ENT), Angular Second Moment (ASM), and Correlation (COR).

### 2.5. Maturing Period Calibration and Correction

#### 2.5.1. Maturing Period Calibration

The calibration of the buckwheat maturity period can be realized according to the acquisition time of UAV-RGB images in each period. That is, the number of days between the harvest date and the current image capture date of buckwheat is defined as the maturity period, which can be denoted as $MD_{i}$, and it facilitates the establishment of the later regression model. The harvest date of buckwheat in this experiment was 15 October 2021, and the capture date of buckwheat images in the different periods was 9, 15, 21, 28 September and 8, 12 October 2021, respectively; therefore, the corresponding interval time of 36 days, 30 days, 24 days, 17 days, 7 days, and 3 days can be preliminarily considered as the buckwheat maturity period, and its physical meaning is the number of days in the maturity period.

#### 2.5.2. Maturing Period Correction

Most buckwheat has matured in the harvest time, but there are still a few immature grains in the calibration process, which will reduce the accuracy of the later model training. Figure 3 shows the images taken at different locations on the same date of buckwheat maturity. It can be seen in Figure 3a that mature grains account for a relatively small proportion at the end of the flowering period; In Figure 3b, most of the grains are mature, and the overall color is brown. Therefore, it is necessary to recalibrate the maturity period according to the color characteristics of the actual canopy in each buckwheat plot. In order to reduce the difficulty of calibration, it is generally considered that the error before and after the actual maturity of buckwheat in the harvest time should not exceed three days. In the process of calibration, the proportion of brown pixels can be approximately regarded as the maturity index, which can be marked as $D_{i}$, and the value range is (−1.5, 1.5); hence, the revised maturity date is $MMD_{i}$ as shown in Formula (2).
(2)$$MMD_{i}=MD_{i}+D_{i}$$

### 2.6. Feature Selection

In order to explore features that can effectively represent the maturity information of buckwheat, this paper extracts vegetation indexes, texture features, and color features of the UAV-RGB image to initialize the feature space. Feature selection can obtain the best combination of feature vectors to make the prediction results the best and ensure the prediction accuracy of the model while reducing the amount of calculation and the difficulty of model learning. The Pearson correlation coefficient can effectively measure the linear correlation among feature vectors and effectively reduce the redundancy of the feature space by removing feature vectors with a high correlation coefficient. Therefore, this paper uses the Pearson correlation coefficient for feature space analysis.

When performing regression analysis on the optimized feature space, there is a relationship of rising first and falling later between the dimension of the feature vectors and the prediction result of the model; thus, the dimension of the feature vectors will seriously affect the prediction accuracy of the model. Therefore, it is necessary to search the optimal dimensions and combinations of feature vectors that make the prediction results the best. Encapsulated feature selection integrates a regression model into the process of feature selection, takes the cross-validation results as evaluation criteria, and selects feature vectors with high contribution values as the final results. Common encapsulated feature selection methods include stability selection, sequential feature selection, and recursive feature elimination [26]. In this paper, recursive feature elimination (RFE) was used to select the optimal combinations of feature vectors, and its working principle is to select features by continuously reducing the volume of the feature sets through recursive methods. The implementation steps of the algorithm are as follows:(1)The initial feature space was the combination of all feature vectors, through which the regression model was trained. The importance of each feature was determined by the attributes of the correlation coefficient and feature importance;(2)The features with the lowest importance were removed from the current feature combinations, and then the process of feature pruning was repeated recursively until the set number of features was reached;(3)Since the recursive feature elimination method needs to manually set the number of features, it is unable to automatically determine the optimal number of features. Therefore, cross-validation can be introduced into the recursive feature elimination, that is, recursive feature elimination with cross-validation (RFECV) [27]. It can automatically determine the optimal number and combinations of features by using the regression model in the process of cross-validation, making the prediction results of the model optimal. In the process of FRFECV, this paper used five-fold cross-validation to select the number and combinations of features.

### 2.7. Regression Model Establishment

The processes of feature extraction, model establishment, and data analysis of the UAV-RGB buckwheat images was realized in Python 3.8.8. The computer is configured as Inter (R) Core (TM) i7-6700@3.4 GHz, and the memory is 8 G. In this paper, decision tree regression, linear regression, random forest regression, AdaBoost regression, gradient boosting regression, and extra tree regression [28] were used for comparative experiments to verify the application of UAV-RGB images in predicting buckwheat maturity. The above regression models were integrated into the process of FRFECV to obtain the optimal dimensions and combinations of feature vectors.

### 2.8. Model Evaluation Indexes

In order to effectively evaluate the importance of each feature vector in regression analysis, the permutation feature importance (PFI) index was introduced. It indicates the decrease of score in the regression model when the value of a single feature vector is randomly disturbed. The PFI score represents the dependence degree of the model on a feature vector, and the prediction performance of different features in model training can be determined by the sorting result of PFI. The Coefficients of determination ($R^{2}$) and Root Mean Square Error (RMSE) [29] were used to verify the prediction accuracy of the model. The formulas are as follows (3)–(5): (3)$$PFI_{j}=s-\frac{1}{K}\sum_{k=1}^{K}s_{k,j}$$
where $PFI_{j}$ represents the importance of the arrangement characteristics corresponding to the jth feature vector; *S* is the reference value of a specific regression model after feature vector learning, and the reference value of the regression model is calculated by $R^{2}$ of the model; K indicates the number of times that the feature vector value is scrambled.
(4)$$R^{2}=\frac{\sum_{i=0}^{n}{\left(X_{i}-\bar{X}\right)}^{2}{\left(Y_{i}-\bar{Y}\right)}^{2}}{n\sum_{i=0}^{n}{\left(X_{i}-\bar{X}\right)}^{2}\sum_{i=0}^{n}{\left(Y_{i}-\bar{Y}\right)}^{2}}$$
(5)$$RMSE=\sqrt{\frac{1}{n}\sum_{i=0}^{n}{\left(Y_{i}-X_{i}\right)}^{2}}$$
where $X_{i}$ and $Y_{i}$, respectively, represent the estimated and measured values of the days from maturity; $\bar{X}$ and $\bar{Y}$ represent the mean value of $X_{i}$ and $Y_{i}$; n is the number of samples.

## 3. Results

### 3.1. Feature Selection Results

The Pearson correlation coefficient method was used to analyze the 22 extracted feature vectors. Figure 4 is the thermodynamic diagram of the correlation coefficient feature vectors, and the darker the color is, the higher the correlation is between feature vectors. It can be seen from Figure 4 that the correlation coefficient between HSV_H and HLS_H is 1, as the calculation formulas of component Hue in HSV and HLS are the same, then feature HSV_H can be removed. The correlation coefficients between GLA and ExG, HSV_S, Lab_b are 0.996, 0.933 and 0.941, respectively, and their correlation coefficients all exceed 0.9, as their contribution in regression analysis are similar; thus, Lab_b, ExG and HSV_ S can be removed and GLA retained. Similarly, 7 feature vectors were removed finally, and the original 22 feature vectors were reduced to 15: vegetation indexes (NGRDI, GLA, VARI), color features (Lab_a, HLS_H, HLS_L, HLS_S, HSV_S, HSV_V), texture features (LBP, Gabor, HOM, CON, COR, ASM).

RFECV was used to twice optimize the feature space. In the process of cross-validation, six regression models of decision tree regression, linear regression et al. were embedded in RFE, and the relationship between the regression prediction values and feature vectors was used for optimal selection. Figure 5 shows the optimal number of feature vectors corresponding to different regression models, in which the abscissa is the dimension of feature vectors, and the ordinate is the cross-validation accuracy of the prediction models. It can be seen from Figure 5 that when the number of the selected features exceeds a certain value, the prediction results of all models tend to be stable, and the increase of the number of feature vectors will not improve the accuracy of the models, but will greatly increase the amount of computation. Therefore, it is unreasonable to blindly increase the number of feature vectors.

In the decision tree regression model, when the number of feature vectors reaches 2, the prediction result is optimal, and the corresponding feature vectors are color feature (HLS_S) and texture feature (ASM), and the feature space is defined as “1”. In the linear regression model, the prediction result is optimal when the number of feature vectors reaches 4, and the corresponding feature vectors are vegetation indexes (NGRDI, VARI) and texture features (ASM, COR), and it is defined as feature space “2”. In the random forest regression model, when the number of feature vectors reaches 3 after feature selection by RFECV, the result is optimal, and the corresponding feature vectors are texture features (ASM, COR) and color feature (HLS_S), and it is defined as feature space “3”. In the AdaBoost regression model, the optimal feature vector dimension is 4, which are texture features (CON, ASM and COR) and color feature (HLS_S), and it is defined as feature space “4”. In the gradient lifting regression model, when the number of feature vectors reaches 5, it reaches the optimum, which is defined as feature space “5”, and the feature space includes vegetation index (VARI), texture features (CON, ASM and COR) and color feature (HLS_S). The extreme random tree regression model needs 10 feature vectors to reach the optimum, which is defined as feature space “6”, and it includes vegetation index (NGRDI), texture features (CON, ASM, COR, HOMO, Gabor) and color features (HLS_L, HLS_H, HLS_S, Lab_a).

After feature selection by Pearson correlation coefficient and RFECV, the optimal dimensions and combinations of feature vectors that correspond to different regression models were obtained finally. Figure 6 shows the PFI values of feature vectors corresponding to different regression models, in which the abscissa is the importance parameter of the arrangement characteristics, and the ordinate is the feature vectors. The boxplot can reflect the distribution of different feature vectors on PFI values. It can be seen from Figure 6 that the importance of feature vector PFIs are different in the six regression models. The importance of the arrangement characteristics of texture feature COR in decision tree regression, random forest regression, AdaBoost regression, and gradient lifting regression are high. The texture features of LBP and Gabor are difficult to reflect the texture characteristics of the whole image on a single value, and the importance is much lower in all six regression models. The experimental results show that the importance of the permutation characteristics of each feature vector is basically consistent with the feature space selected by RFECV, which verifies the effectiveness of our method.

### 3.2. Prediction Algorithm Selection Results

After feature selection, feature vector optimization, and combination, the optimal feature vector combinations for each model were obtained. In order to evaluate the matching results between the regression models and the feature spaces, vegetation indexes, texture features, color features, and all features, respectively, combined with different regression models were conducted for contrast experiments. The regression analysis models with the prediction results of the corresponding feature space are shown in Table 2, and the results are the average of several experiments.

According to Table 2, although the prediction results of the six regression models are good for all feature spaces, they are not optimal compared with other feature spaces, which indicates that more feature vectors does not mean better. It is necessary to find the optimal feature space for different regression models. The prediction results of the vegetation index in the six regression models are all poor; hence, it can be considered that the vegetation index alone cannot accurately predict buckwheat maturity. Compared with the vegetation index, the regression results of different regression models are better for texture features and color features. In the decision tree regression model, the feature space with the highest $R^{2}$ is “1”, and its RMSE is 2.27; the regression prediction result is the best compared with other model spaces, which is the same as the optimization result of RFECV. In the linear regression model, its optimal space is feature space “2”. There is a slight decrease in $R^{2}$ and RMSE compared with all feature spaces, but it can greatly reduce the calculation amount of regression analysis. To summarize, the prediction results of random forest regression in feature space “3”, “4”, and “5”are the same, and $R^{2}$ and RMSE are the highest compared with other feature spaces. Gradient lifting regression performs best in the same feature space, and the result of feature space “5” is the best compared with other feature spaces. Therefore, the method combining feature space “5” with gradient lifting regression can be used to realize the prediction of buckwheat maturity based on UAV-RGB images. The optimal determination coefficient ($R^{2}$) and root mean square error (RMSE) are 0.981 and 1.70, respectively, and the results basically meet the results of RFECV optimization.

## 4. Discussion

In this paper, the Pearson correlation coefficient was used for preliminary dimensionality reduction of the feature space. We reduced the original 22 feature vectors to 15, proving the effectiveness of the Pearson correlation coefficient, which included vegetation indexes (NGRDI, GLA, VARI), color features (Lab_a, HLS_H, HLS_L, HLS_S, HSV_S, HSV_V), and texture features (LBP, Gabor, HOM, CON, COR, ASM). 

Regression models were integrated into the process of feature selection, in which the cross-validation results were used as the evaluation criteria, and the feature vectors with higher contribution values were selected as the results. Through RFECV, the optimal number and combinations of feature vectors corresponding to different regression models were obtained. Moreover, six feature spaces were defined. The experimental results show that the method of combining feature space “5” (vegetation index VARI; texture features CON, ASM and COR; and color feature HLS_S) with gradient lifting regression can realize the prediction of buckwheat maturity, and its $R^{2}$ and RMSE are 0.981 and 1.70, respectively, when achieving the best result in all the combinations. 

Most of the images used for crop maturity predictions are satellite remote sensing images [30,31] (research on rice and corn) and UAV images. As there has been little previous work applied to crop maturity predictions in UAV images, let alone the applications in buckwheat, this work can be compared to the prediction results for soybean. Two complementary convolutional neural networks (CNN) were developed to predict the maturity date in reference [10], and the data were acquired from three growing seasons in the USA and Brazil, and the RMSE was 2.0 days. In reference [11], the prediction of soybean maturity dates was realized using UAV multispectral imagery, the experiment was conducted at an experimental field at the University of Missouri, Novelty, Missouri, United States, and the $R^{2}$ and RMSE were 0.81 and 1.4 days, respectively. Compared with state-of-arts on soybean, in this paper, the improvement of $R^{2}$ is obvious and the value of RMSE can meet the general error requirement. Although the use of deep learning can perform well in predictions, it must depend on a large amount of data to guarantee the robustness of the model. The regression analysis model used in this paper can also meet the prediction requirements with small data sets. Furthermore, the optimal feature spaces corresponding to different regression analysis methods can be obtained automatically. Moreover, when conducting feature dimension reduction with PCA (Principal Component Analysis), the features obtained usually have no practical significance, while the optimal feature space obtained in our method has practical physical significance that can represent the characteristics of buckwheat maturity. In addition, it can also reduce the amount of calculation for feature extraction in later detection.

Analyzing the optimal feature space showed the optimal feature space with different regression models contains texture features (ASM and COR) and color features (HLS_S), and the vegetation index performs poorly. Therefore, the correlation between texture features, color features, and maturity can be further explored. Remote sensing technology can only obtain the phenotypic information of crops, and it is difficult to describe the internal mechanism changes of crops. In reference [32], visible and near-infrared hyperspectral imaging was employed to evaluate the maturity stage and moisture content of fresh okra fruit precisely, and the physicochemical analysis indicated that there was a negative correlation between maturity and moisture content. To assess the viability of growing cover crops in Denmark [33], a phenology model was developed and applied to predict the harvest date of spring barley and winter wheat. Therefore, the phenological information, crop growth information, and other chemical indicators can be introduced to the prediction model of crop maturity in the future.

## 5. Conclusions

In this study, RFECV was used to integrate six regression models of decision tree regression and random forest regression et al. into feature selection, and the optimal dimension of the feature vector was obtained during the process of cross-validation, and the feature vectors that can effectively predict the maturity of buckwheat in UAV-RGB images were mined in depth. The traditional prediction of buckwheat maturity usually depends on manual judgment to determine the corresponding harvest (sectional harvest and one-time harvest). However, when the planting area is larger, it is difficult to evaluate the maturity of the field manually. Then it is valuable to use UAV to predict the maturity of buckwheat. Moreover, the cost of acquiring and processing images based on RGB and UAV in this paper is relatively low, which can be widely promoted.

For off-line prediction of crops maturity based on UAV images, priority should be given to improving the performance of the regression models, while calculation requirements need not be considered to some extent. However, for on-line prediction, priority should be given to calculation requirements, and the stability of the system needs to be guaranteed. The premise of reducing the amount of computation is to ensure that the performance of the model will not be greatly affected, and finally to provide a theoretical basis for the application of UAV in agricultural on-line detection. Furthermore, the time series images were collected at a fixed flight altitude of the same field. The impact of different flight altitudes and imaging resolutions on the experimental results were not considered. The follow-up work can continue to carry out in-depth mining of the UAV-RGB images, taking into account the spatial location information and weather information. Moreover, it is valuable and necessary for further research on buckwheat phenotype analysis, such as chlorophyll content, nitrogen content, and yield estimation.

## Figures and Tables

**Figure 1 plants-11-03257-f001:**
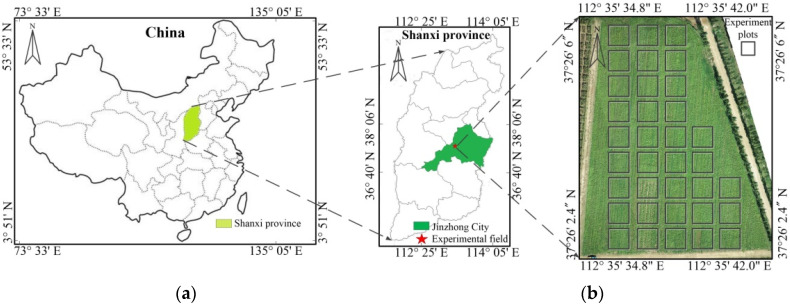
Geographical location of the research area and the buckwheat experimental plots. (**a**) Geographical location of the research area. (**b**) Buckwheat experimental plots.

**Figure 2 plants-11-03257-f002:**
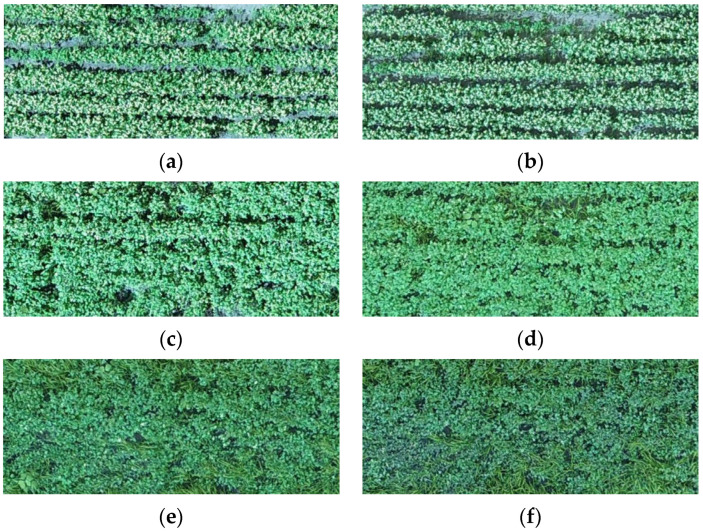
UAV-RGB images of buckwheat in different periods in the same plot. (**a**) Buckwheat squaring period (9 September 2021). (**b**) Buckwheat squaring period (15 September 2021). (**c**) Buckwheat flowering period (21 September 2021). (**d**) Buckwheat flowering period (28 September 2021). (**e**) Buckwheat growth period (8 October 2021). (**f**) Buckwheat maturity period (12 October 2021).

**Figure 3 plants-11-03257-f003:**
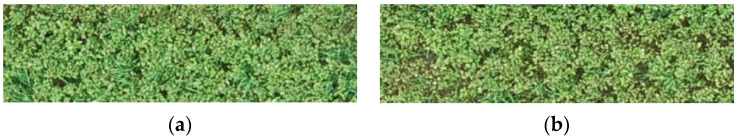
Images taken at different locations on the same date during the buckwheat maturity period. (**a**) Partial immature. (**b**) Basically mature.

**Figure 4 plants-11-03257-f004:**
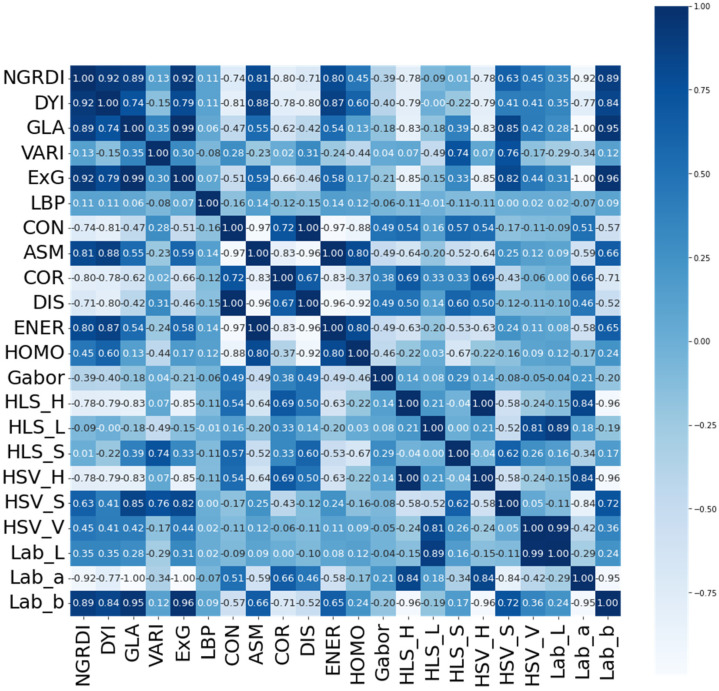
Thermodynamic diagram of the correlation coefficient feature vectors.

**Figure 5 plants-11-03257-f005:**
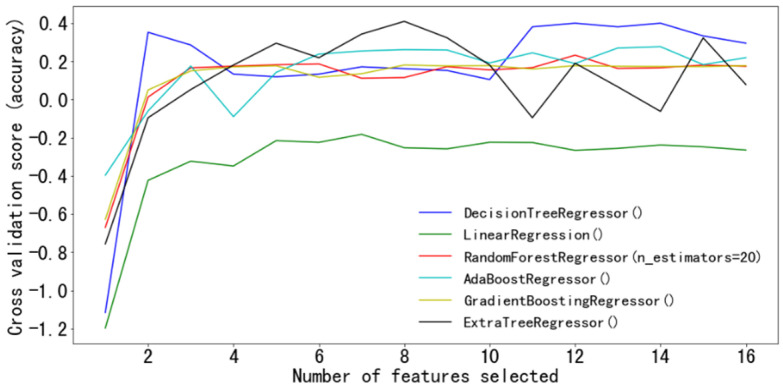
The optimal number of corresponding feature vectors for different regression models.

**Figure 6 plants-11-03257-f006:**
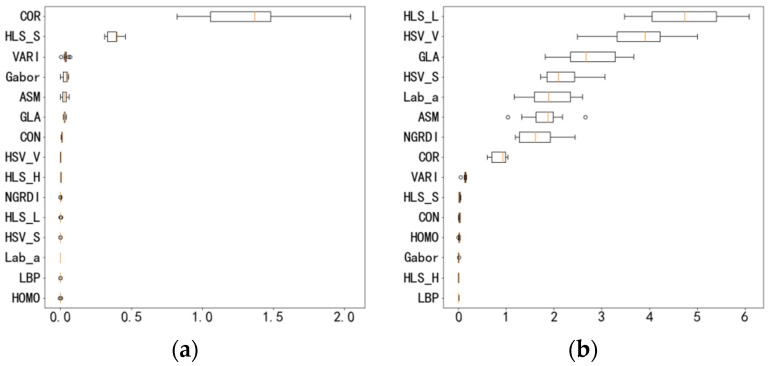
Feature vector PFI values of different regression models. (**a**) Decision tree regression. (**b**) Linear regression. (**c**) Random forest regression. (**d**) AdaBoost regression. (**e**) Gradient lifting regression. (**f**) Extreme random tree regression.

**Table 1 plants-11-03257-t001:** 5 Vegetation indices.

Vegetation Indexes Name	Abbreviation	Formula
Normalized green red difference index	NGRDI	(g−r)/(g+r) ^1^
Green Leaf Algorithm	GLA	(2∗g−r−b)/(2∗g+r+b)
Visible Atmospherically Resistant Index	VARI	(g−r)/(g+r−b)
Excess green index	ExG	2∗g−r−b
Normalized Difference Yellow Index	NDYI	(g−b)/(g+b)

^1^ Note: R, G, and B, respectively, represent the reflection values of the R, G, and B channels of the UAV images.

**Table 2 plants-11-03257-t002:** Regression analysis model and corresponding feature space prediction results.

**Feature Spaces**	Decision Tree Regression		Linear Regression		Random Forest Regression		AdaBoost Regression		Gradient Lifting Regression		Extreme Random Tree Regression	
	$R^{2}$	RMSE	$R^{2}$	RMSE	$R^{2}$	RMSE	$R^{2}$	RMSE	$R^{2}$	RMSE	$R^{2}$	RMSE
Vegetation Indexes	0.354	9.56	0.624	7.30	0.612	7.42	0.683	6.69	0.687	6.66	0.616	7.34
Texture Features	0.869	4.31	0.856	4.52	0.868	4.33	0.907	3.63	0.892	3.91	0.842	4.71
Color Features	0.922	3.27	0.851	4.59	0.848	4.64	0.928	3.17	0.937	2.98	0.792	5.33
Feature Space 1	**0.964**	**2.27**	0.745	6.01	0.949	2.68	0.955	2.52	0.962	2.33	0.929	3.16
Feature Space 2	0.882	4.08	**0.902**	**3.73**	0.868	4.32	0.919	3.38	0.921	3.35	0.867	4.27
Feature Space 3	0.951	2.64	0.855	4.54	**0.949**	**2.68**	0.960	2.39	0.971	2.02	0.924	3.00
Feature Space 4	0.960	2.35	0.856	4.53	0.949	2.68	**0.965**	**2.21**	0.977	1.80	0.927	2.92
Feature Space 5	0.949	2.68	0.855	4.54	0.949	2.68	0.959	2.40	**0.981**	**1.70**	0.926	3.21
Feature Space 6	0.959	2.39	0.887	3.99	0.948	2.71	0.957	2.47	0.975	1.90	0.925	3.21
All Feature Space	0.954	2.54	0.934	3.06	0.948	2.72	**0.966**	**2.20**	0.980	1.70	**0.928**	**3.08**

## Data Availability

The data presented in this study are available on request from the corresponding author.

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
