# Peer review of "Prediction of Buckwheat Maturity in UAV-RGB Images Based on Recursive Feature Elimination Cross-Validation: A Case Study in Jinzhong, Northern China"

_plants, 2022, doi:10.3390/plants11233257_

Round 1
Reviewer 1 Report
General comments
This study utilizes UAV imagery and regression models to predict buckwheat maturity. Although this has applied to other crops, there has been little previous work applied to buckwheat and there is value in being able to better predict maturity to optimize yield and profitability. The paper is generally well-written, though would benefit from additional discussion to further clarify what was learned from this paper relative to prior research. Do different models tend to under or over predict maturity level? How much does this method improve accuracy of maturity estimation relative to previous methods? Any assessment of the value to buckwheat farmers from having this information relative to the cost of acquiring and processing UAV data?
Specific comments
p.4, lines 140 and 148. Missing reference errors. There are others in the paper as well.
p.4, lines 153-156. List in text leaves out the fifth index, the excess green index.
p.5, lines 183-193. This paragraph provides an important rationale for the importance of accurate and rapid assessment of buckwheat maturity. I think it would be preferable to have this discussion earlier in the paper to help make clear the importance of this work.
p.11, 380-381. How valuable is this reduction in calculation amount relative to the reduction in fit? How are you assessing the tradeoff between model performance and calculation requirements?
Reviewer 2 Report
This paper shows a theme in UAV-RGB Images in China. The contribution is significant to the advancement of knowledge. However, some points need to be better detailed for a complete understanding.
Title: It's generic… no country or region indication.
Abstract: It is suggested include more data and information from the study. Example, geographical area/year of the study, ...
Introduction: is comprehensive whit a good overview of problem in context. However, I suggest to describe the applications in studies of agriculture, the objective more directly in final and remove “The remainder of this study is organized as follows. The proposed methods are introduced in detail in Section 2. Then, Section 3 describes experiments conducted with the proposed method on the RGB-UAV remote sensing images of buckwheat and presents comparisons with other regression models in order to verify the effectiveness. Section 4 discusses the performance of our method. Conclusions are presented in Section 5.”
Methods and Data: the method description is good. In Figure 1, include the province's position in China. In page 5, need to adjust:“By analyzing the common color space HSV Error! Reference source not found., HLS Error! Reference source not found. and Lab Error! Reference source not found”.Results: is correctly interpreted. It is necessary to improve the visual resolution of figure 4.
Discussions: It's very superficial. Including more details with other aspects (authors/studies) in the discussions – compare the data obtained with other possible regions in China or countries. Comment more on possible limitations of the use UAV during the field survey.
Conclusions: topics are focused on the objective.
Reviewer 3 Report
Manuscript is well organized and well written. Manuscript is a good scientific contribution to science of image processing and its application in agriculture.
However, there are some minor issues related to unwanted text such as “Error! Reference source not found.” At several places across the manuscript.
I believe this manuscript should be accepted for publication.
Round 2
Reviewer 2 Report
Adjust, remove a duplicated part of each figure (1 and 4).
I still think that the discussion and references/authors should be improved - 4.1 and 4.2
